# Molecular and Biologic Effects of Platelet-Rich Plasma (PRP) in Ligament and Tendon Healing and Regeneration: A Systematic Review

**DOI:** 10.3390/ijms24032744

**Published:** 2023-02-01

**Authors:** Byron Chalidis, Panagiotis Givissis, Pericles Papadopoulos, Charalampos Pitsilos

**Affiliations:** 11st Orthopaedic Department, Aristotle University of Thessaloniki, 57010 Thessaloniki, Greece; 22nd Orthopaedic Department, Aristotle University of Thessaloniki, 54635 Thessaloniki, Greece

**Keywords:** PRP, platelet rich plasma, ligament, tendon, biology, collagen

## Abstract

Platelet-rich plasma (PRP) has been introduced and applied to a wide spectrum of acute and chronic ligament and tendon pathologic conditions. Although the biological effect of PRP has been studied thoroughly in both animal and human studies, there is no consensus so far on the exact mechanism of its action as well as the optimal timing and dosage of its application. Therefore, we conducted a systematic review aiming to evaluate the molecular effect of the administration of PRP in tendoligamentous injuries and degenerative diseases. The literature search revealed 36 in vitro and in vivo studies examining the healing and remodeling response of animal and human ligament or tendon tissues to PRP. Platelet-rich plasma added in the culture media was highly associated with increased cell proliferation, migration, viability and total collagen production of both ligament- and tendon-derived cells in in vitro studies, which was further confirmed by the upregulation of collagen gene expression. In vivo studies correlated the PRP with higher fibroblastic anabolic activity, including increased cellularity, collagen production and vascularity of ligament tissue. Similarly, greater metabolic response of tenocytes along with the acceleration of the healing process in the setting of a tendon tear were noticed after PRP application, particularly between the third and fourth week after treatment. However, some studies demonstrated that PRP had no or even negative effect on tendon and ligament regeneration. This controversy is mainly related to the variable processes and methodologies of preparation of PRP, necessitating standardized protocols for both investigation and ap-plication.

## 1. Introduction

Both ligament and tendon tissues are formed mainly of water and collagen (COL) type I [1,2]. Ligaments are also composed of other COL types (III, VI, V, XI and XIV), fibroblasts, proteoglycans, elastin and proteins, such as actin, laminin and integrins [2]. Important structural components of tendons are COL type III, V and XI, tenocytes, elastin, proteoglycans (e.g., decorin, lubricin and versican), glycoproteins [e.g., cartilage oligomeric matrix protein (COMP) and tenascin-C], and matrix metalloproteinases (MMP) [3]. Collagen type I is predominant in mature ligaments and tendons and exhibits higher strain-resistance properties. In contrast, collagen type III is found in greater abundance in the developing and healing tissue and shows decreased tensile strength [4]. Ligament fibroblasts and tenocytes are the first line of defense to injury or repetitive mechanical stress and subsequent degeneration. As a result, they can alter the extracellular matrix (ECM) gene and protein expression [5]. Proteoglycans bind to collagen fibers and contribute to collagen cross-linking [6]. Elastin, which is more prevalent in ligaments than tendons, is an ECM protein that offers fatigue-resistance and regulates energy storage [7].

The injury of tendoligamentous tissues is followed by a three phased of healing process: hemorrhage with inflammation, matrix and cellular proliferation, and remodeling and maturation [8]. The repair phase includes the production of a disorganized matrix with randomly aligned collagen fibers, mainly of COL type III, leading to a weaker structure prone to rupture and tear [2,9]. This is followed by the remodeling phase, which may last up to 1 year, and is characterized by the gradual replacement of disoriented COL type III from organized COL type I that creates a more competent tissue construct [10]. In the setting of tendinopathy, there is increased COL type III synthesis that outnumbers COL type I, aberrant ECM organization, abnormal vascularity, and modification of protein composition and content. These changes result in decreased tendon tensile strength and ability to withstand mechanical loads [11,12]. Apart from the local biological response, reduction of loading and shearing forces at the injury site is also an important factor for facilitating tendoligamentous tissue healing [13,14].

Tendon and ligament injuries and degeneration may be associated with poor healing potential and high incidence of recurrence due to the low metabolic rate of tendons and ligaments alike [15]. Platelets are the “first responders” after wounding, which triggers their activation and aggregation [16]. Once activated, platelets release many growth factors that could affect hemostasis [cyclo-oxygenase, adenosine diphosphate, von Willebrand factor (vWF), fibrinogen and P-selectin], the regulation of inflammation [interleukin (IL)-1β and platelet factor 4], angiogenesis [epidermal growth factor (EGF), hepatocyte growth factor (HGF), MMPs, platelet-derived growth factor (PDGF) and vascular endothelial growth factor (VEGF)], and tissue remodeling [stromal cell-derived factor 1 and basic fibroblast growth factor (FGF)] [17]. Clinical studies have shown that platelet derived growth factors, contained in platelet rich plasma (PRP), may be beneficial for many pathological conditions and augment tendon and ligament repair in both traumatic and degenerative diseases [18,19,20,21,22,23]. They stimulate the cell surface receptors and intracellular signaling proteins that control both the repair and remodeling mechanisms [24,25,26,27,28,29]. Furthermore, PRP can induce the differentiation of tendon stem/progenitor cells (TSCs) into active tenocytes and subsequently produce abundant collagen to promote the healing of injured tendons [30,31].

Although the biological effect of PRP has been studied thoroughly in both animal and human studies, there is no consensus about the exact mechanism of action as well as the optimal timing and dosage of its application. The aim of the current systematic review is to present the in vitro and in vivo molecular effect of the administration of PRP in tendon and ligament injuries and diseases and provide the current evidence-based recommendations in clinical practice.

## 2. Materials and Methods

A systematic review of the literature utilized the Medline database regarding the effect of PRP on ligament and tendon healing and regeneration. The inclusion criteria included comparative studies in the English language concerning the biological and molecular effect of PRP on ligament and tendon healing and metabolism. Articles that described the biomechanical or functional outcome of PRP without reference to biology, examined the effect of PRP after anterior cruciate ligament (ACL) reconstruction, did not compare the results with controls and included case series or case reports were excluded from further evaluation.

The following keywords were applied: “PRP ligament”, “Platelet-rich plasma ligament”, “PRP tendon”, and “Platelet-rich plasma tendon”. The initial data search identified 2839 articles. After duplicate removal and title and abstract screening following the inclusion and exclusion criteria, 103 articles were considered eligible for full-text retrieval. Finally, and after detailed evaluation of the full texts, 48 articles were included for further analysis. Specifically, 24 of them referred to ligaments, 21 to tendons and 3 to both tendons and ligaments (Figure 1).

The data from each study were entered into a Microsoft Excel database and analyzed using the SPSS computer program (IBM Corp, 2017. IBM SPSS Statistics for Windows, Version 25.0. Armonk, NY: IBM Corp.). Continuous variables were reported as means with standard deviations or medians. Categorical variables were presented as proportions. Sub-group column proportions were compared for categorical variables using z-test and Bonferroni correction was used to adjust for *p* values. All tests were two-sided and statistical significance was assumed at a *p* value of less than 0.05.

## 3. Results

### 3.1. Effect of PRP on Ligament Tissue

#### 3.1.1. In Vitro Studies

Thirteen studies (eight animal and five human) examined the correlation of PRP with the upregulation of ligament anabolic activity using incubating cultures of ligament fibroblasts in PRP-containing matrices [32,33,34,35,36,37,38,39,40,41,42,43,44]. In animal studies, the most common cell sources were the bovine or rabbit ACL (*n* = 5) and the equine suspensory ligament (*n* = 3), while in human studies the ACL was the main material used (*n* = 3), followed by the periodontal ligament (*n* = 2). Cultures were studied for a mean time of 8 days (range: 6 h-21 days). The outcomes focused on the histological cell changes, the expression of ligament metabolic genes (COL1, COL3, COMP, MMP3, MMP13 and decorin), and the release of growth factors associated with ligament healing and extracellular matrix composition [VEGF, Thrombospondin-1 (TSP-1) and HGF]. According to the histological findings, PRP was associated with increased cell viability (*p* < 0.05), proliferation (*p* < 0.05) and migration (*p* < 0.05), and decreased apoptotic rate (*p* < 0.05). Additionally, the gene expression of COL1 (*p* < 0.05) and COMP (*p* < 0.05) was upregulated after fibroblast exposure to PRP compared to controls (Table 1).

##### Animal Studies

Smith et al. [41] cultured equine suspensory ligament cells for 48 h in seven different matrices and compared the effect of PRP and acellular bone marrow on fibroblast protein production. A higher expression of COMP and the incorporation of H-Leucine were noticed after the addition of PRP in the culture media, indicating overall protein increased synthesis. Moreover, acellular bone marrow evoked an even higher production of these molecules.

Schnabel et al. [40] cultured equine suspensory ligament fibroblasts in plasma, whole blood, bone marrow aspirate (BMA), platelet poor plasma (PPP) and PRP for 3 days. The authors associated the presence of PRP in culture media with increased COMP and MMP3, and decreased COL3A1, MMP13, COL1A1, COL1A1:COL3A1 and decorin gene expression compared to 10% plasma matrix (control group).

McCarrel et al. [38] cultured equine suspensory ligament samples for 4 days in matrices containing PPP, PRP, BMA, or none of them. Compared to all modalities, PRP demonstrated the greatest upregulation of COL1A1, COMP, decorin and MMP3 genes with the simultaneous decrease of COL3A1 and MMP13 gene expression. The ratio of COL1A1: COL3A1 was also increased in this group.

Cheng et al. [34] cultured porcine ACL cells in collagen type I hydrogel loaded with platelets, PPP or PRP, for 14 days. An increased cellular metabolic activity and reduced apoptotic rate were found in all the agent-based scaffolds. The histological evaluation revealed the most mature appearance in the PRP group, where the cells were elongated with centrally positioned elongated nuclei; they were oriented along the longitudinal axis of the structure and they had the highest average nuclear aspect ratio. Furthermore, hydrogels in the PRP group showed a wave fiber-like structure of collagenous extracellular matrix aligned with the longitudinal axis of the constructs. The collagen fibrils were abundant in the matrix, and most of them were aligned with the cells and packed into collagen fibers. Additionally, PRP was associated with increased COL1A1 and COL3A1 gene expression at the highest rate.

In another study from the same authors [33], ACL cells from immature, adolescent and adult pigs were cultured in COL hydrogel with or without PRP and the correlation between age and metabolic stimulation of PRP was examined. Higher cellular metabolic activity and reduced apoptotic rate were found in immature and adolescent animals after the administration of PRP. Furthermore, the upregulation of COL1A1 and COL3A1 gene expression was detected in the same PRP and age groups.

Yoshida et al. [44] cultured fibroblasts derived from pig ACL in six matrices containing PPP, PRP and peripheral blood mononuclear cells (PBMC) in different combinations. After 14 days, the presence of PRP and PBMC in the culture media was related to increased cell metabolic activity, better collagen organization, higher procollagen type I and type III gene expression and increased IL-6 production.

In another study, Yoshida et al. [42] cultured porcine ACL fibroblasts for 14 days in collagen matrices that contained PPP or three different concentrations of PRP or nothing. In respect to histological findings, increased cell metabolic activity and decreased apoptosis was found in all PRP groups, but the optimum amount and concentration of PRP couldn’t be defined. In contrast, the expression of COL1A1 and COL3A1 genes was higher in the PRP groups and was also inversely proportional to the amount of PRP added.

In 72-h cultures of fibroblasts derived from rabbit ACL, Zheng et al. [43] studied the single and synergistic effect of PRP with Sanguisorba officinalis L. polysaccharide. Increased cell viability and migration as well as decreased apoptosis were evident after introduction of PRP in the culture plate. Furthermore, PRP increased the expression of runt-related transcription factor 2 (Runx2), alkaline phosphatase, bone morphogenetic protein (BMP) 2, collagen type I and osteoprotegerin proteins compared to controls. The authors also noticed a negative effect of PRP on toll-like receptor 4 (TLR-4)/nuclear transcription factor κB pathway that verified by the inhibition of TLR-4 and p65 phosphorylation. Finally, the addition of Sanguisorba officinalis L. polysaccharide considerably enhanced the action and biologic activity of PRP.

##### Human Studies

Fallouh et al. [36] cultured cells from removed ACL remnant tissues during ACL reconstruction on matrices containing PPP, 5% PRP, 10% PRP, or nothing. At the fourth day, cell viability was higher in the 10% PRP group. After 7 days, cells in PRP-cultures produced more total collagen and demonstrated the overexpression of the COL3 gene. However, the expression of the COL1 gene and the COL/μg DNA was similar in all culture groups.

In 2-day cultures of human ACL cells, Dhillon et al. [35] found increased cell viability and proliferation, but the same incidence of apoptosis after exposure to 5% PPP, 5% PRP or 10% PRP compared to control cultures. Moreover, all PRP concentrations performed equally, and no statistically significant difference was identified.

Krismer et al. [37] studied the effect of PRP with different concentrations of leucocytes on human ACL fibroblasts cultured for 21 days. Independently of the presence of leucocytes, PRP was correlated with increased cell proliferation but not extracellular matrix production. In order to address the effect of PRP on the overall fibroblast metabolism, the authors quantified the expression of genes that associated with anabolic and catabolic activity. They found increased COL1A2, COL2A1, tenascin C (TNC), scleraxis bHLH transcription factor, tenomodulin (TNMD), aggrecan, mohawk homeobox and MMP13 and decreased COL3A1, MMP3 gene expression in the leukocyte rich platelet rich plasma (LR-PRP) group compared to leukocyte poor platelet rich plasma (LP-PRP) or non-PRP matrices groups. However, only the differences in MMP13 and MMP3 levels reached statistical significance.

In a 3-day culture of human periodontal ligament cells, which were simulated or not with PRP, Anitua et al. [32] found that the presence of PRP was associated with greater cell proliferation, migration and adhesion to collagen type I. Furthermore, they measured and compared the synthesis of some biomolecules and extracellular matrix components by using the ELISA method. In the PRP group, the concentrations of VEGF, TSP-1, HGF, connective tissue growth factor (CTGF), HGF and procollagen type I were increased. On the contrary, the release of α2 integrin was lower compared to the non-PRP sample group.

In a similar study, Rattanasuwan et al. [39] noticed that the addition of 10% PRP in culture media improved the proliferation, migration and attachment of fibroblasts derived from human periodontal ligaments when compared to controls.

#### 3.1.2. In Vivo Studies

Thirteen trials studied the biological effect of PRP on animal ligaments in vivo, either by using PRP-fibrin scaffolds or injecting PRP at the site of ligament injury [45,46,47,48,49,50,51,52,53,54,55,56,57]. The ACL (*n* = 8) and medial collateral ligament (MCL) (*n* = 6) were the most common tissues used, and the results were merely based on histological findings. The mean time of histological examination after PRP application was 8 weeks (range: 2–24 weeks). A relatively common score applied to evaluate the ligament maturity was the ligament tissue maturity index (LTMI), which was based on cellularity, collagen structure and vascularity [46,47,54,55,56]. A higher LTMI score was correlated with PRP application but this difference was not statistically significant (*p* = 0.183). Similar results were also identified for collagen (*p* = 0.095) and vascularity (*p* = 0.235). However, the cellularity score was improved in a statistically significant manner (*p* < 0.05) (Table 2).

##### Animal Studies

Murray et al. [56] studied the effect of collagen-PRP component on dog ACL tear healing. Compared to controls, a more advanced defect filling process was noticed at six weeks after the procedure.

In a similar study in porcine models, the same authors group found that the implantation of a PRP-collagen matrix was associated with hypercellularity, hypervascularity and decreased inflammatory response at the site of ACL rupture after four weeks. [54]

Similarly, Murray et al. [55] identified that extraarticular ligaments healed faster than intraarticular ones after studying MCL, patellar tendon and ACL tears in dog models. Their result was attributed to better soft tissue coverage of extraarticular ligaments, which could respond immediately to injury with cell and vessel recruitment and production of a fibrin-platelet scaffold. The authors also compared the healing process between the intraarticular ACL and the extraarticular MCL after application of collagen-PRP hydrogel at 6 weeks. No difference in LTMI score and similar levels of fibronectin, fibrinogen, PDGF-A, transforming growth factor (TGF)-β1, FGF-2, procollagen type I and vWF growth factors were recorded. This result indicated decrease healing time of the intraarticular ACL that was similar to that of the extraarticular ligament tear.

Joshi et al. [50] examined the effect of the collagen-PRP matrix on repaired ACL in an immature pig model. After 3 months, the PRP group showed increased cellularity of the healing ligament with fusiform-shaped cells and decreased vascularity compared to controls. However, the histological features of the healed tendon were slightly dissimilar to the intact ligament.

In a pig model, Mastrangelo et al. [52] augmented the bilateral ACL repair with a PRP-collagen scaffold having platelet concentration of either five or three times the systemic baseline of platelets. At 13 weeks, fusiform shaped cells were oriented along the length of the ligament in all specimens. The highest PRP load led to greater fibroblastic concentration, more organized collagen, and higher cellularity and vascular subscores of the LTMI. However, the mechanical properties of the repaired ligaments were similar in both groups.

Haus et al. [49] compared the histological findings of ruptured porcine ACL that repaired with or without collagen-PRP composite after 15 weeks. The augmentation group displayed more advanced remodeling at the insertion site, hypercellularity and increased collagen organization at the fibrous zone. Moreover, a relatively large proportion of the collagen oriented perpendicular to the insertion site in a densely packed arrangement. Additionally, there was a distinct fibrocartilage layer with relatively good organization of the collagen perpendicular to the subchondral plate. The authors also reported that the procedure was less mature in the older animals.

Yoshika et al. [57] examined rabbits with ruptured MCL that either were left untreated or treated with PRP clot application and compared the histological findings after 3 and 6 weeks. Cellularity and cellular size were decreased in both groups between the two time points. However, cells and new collagen fibers were more organized and longitudinally aligned within MCL fibers in the PRP group.

Harris et al. [48] examined the effect of PRP on intact rabbit MCL at 2 and 6 weeks after injection. At 2 weeks, they found monocytic and lymphocytic inflammatory infiltration and tendon thickening, while at 6 weeks there was persistent but less prominent inflammation. Normal appearance was found in saline-injected ligaments.

Matsunaga et al. [53] in a rabbit model study removed the MCL and replaced it with a PRP-fibrin scaffold. At 12 weeks, a neo-ligament tissue consisted of dense and longitudinally aligned collagen bundles, and direct fibrocartilage insertions were identified.

In a rat model study, Amar et al. [45] noticed that 3 weeks after injection of PRP at the ruptured MCL site, there was no difference in terms of cellularity, collagen organization and vascularity compared to controls, where saline only was injected. The occurrence of fat cells, inflammatory foci, blood vessels, loose and disorganized collagen was also similar in both groups.

Bozynski et al. [46] compared the outcomes of observation, washout and LP-PRP on dog ACL partial tear healing. At 7 weeks, no difference in myxoid or mucinous degeneration and collagen fiber orientation was identified between the different treatment modalities.

Similarly, Cook et al. [47] studied the effect of LP-PRP on partial ACL tear in a dog model. They applied 5 intraarticular injections of PRP at 1, 2, 3, 6 and 8 weeks and studied the outcomes after 24 weeks. The LP-PRP showed a beneficial effect on tendon healing and tissue quality compared to saline-injected subjects.

LaPrade et al. [51] studied the effect of two different concentrations of PPP and PRP on ruptured MCL rabbit ligaments using the LTMI. At 6 weeks, no modality achieved remodeling and maturity similar to the intact tendon. In addition, the overall scores and subscores for cellularity, collagen and vascularity assessment were equivalent to the saline-treatment group. The authors also reported than while PPP and PRP did not accelerate the healing process, the fourfold amount of PRP, which was the largest concentration injected, had an adverse effect on ligament remodeling and collagen orientation.

### 3.2. Effect of PRP on Tendon Structure

#### 3.2.1. In Vitro Studies

The impact of PRP on tendon cell biology was investigated in ten studies (five animal and five human) that cultured tenocytes in media containing PRP [31,38,58,59,60,61,62,63,64,65,66]. The main sources of tenocytes were the equine flexor digitorum superficialis tendon in animal models (*n* = 2) and the hamstring tendons in humans (*n* = 3) [38,58,59,60,63]. The mean period of cultures was 8 days (range: 3 days-3 weeks). While the results of human studies were based mainly on the histological changes, the animal studies were focused more on gene expression, especially COL1A1, COL3A1, MMP3 and MMP13 [31,38,60,61,62,63,64]. The PRP application led to increased tenocyte viability (*p* < 0.05), proliferation (*p* < 0.05) and collagen production and orientation (*p* < 0.05). Furthermore, higher COL1A1 gene expression (*p* < 0.05) was identified. The expression of COL3A1, COL1A1:COL3A1, MMP3 and MMP13 was also improved without, however, reaching statistical significance (*p* = 0.349, *p* = 0.112, *p* = 0.082 and *p* = 0.187 respectively) (Table 3).

##### Animal Studies

An extensive analysis of the impact of PRP on gene expression was conducted by Hudgens et al. [61] using rat tail tendon fibroblasts that cultured in PRP collagen for 5 days. Regarding the expression of genes associated with tendon growth, PRP upregulated BMP7, downregulated CTGF and insulin-like growth factor (IGF)-I, but did not affect TGF-β. With respect to inflammation- and immune-modulating cytokines, PRP increased the expression of chemokine (C-C motif) ligand (CCL) 2, CCL7, IL-1a, IL-6, IL-10, and tumor necrosis factor alpha, downregulated IL-15 and showed no effect on IL-1b and VEGF. As for the extracellular matrix synthesis and remodeling, no change in the expression of the hyaluronic acid synthase enzymes hyaluronan synthase (HAS) 1 and HAS2 was found. Moreover, a reduced elastin expression along with a slight upregulation of the cross-linking enzyme lipoxygenase (LOX) were observed. The basement membrane COL8 and the proteoglycan lubricin were elevated, while a downregulation of the major fibrillar collagens, COL1 and COL3 as well as of the genes associated with collagen fibril assembly, including cartilage intermediate layer protein, fibromodulin, COL12 and COL14 was reported. Additionally, PRP upregulated the major collagenase MMP13, along with the stromelysins MMP3 and MMP10 and the gelatinase MMP9. However, it did not influence the collagenase MMP8, gelatinase MMP2 and tissue inhibitor of metalloproteinases (TIMP) 1 or TIMP2 genes expression. Concerning fibroblast proliferation, the relevant marker Ki67 was slightly increased. For genes involved in tendon fibroblast specification and differentiation, a downregulation of early growth response protein (EGR) 1, EGR2, scleraxis (SCX) and TNMD was detected. Regarding autophagy, PRP did not affect the expression of autophagy-related protein 10, B-cell lymphoma/adenovirus E1B interacting protein 1, gamma-aminobutyric acid receptor-associated protein–like 2, beclin 1 and Tripartite motif containing 13. Furthermore, the upregulation of FosB, FOS-related antigen 1, and transcription factor Jun, which are transcription factors involved in inflammation, was noticed. Similarly, no change in the deacetylase sirtuin 1 or the nitric oxide–producing inducible nitric oxide synthase gene expression was identified. Prostaglandin production was induced by the upregulation of phospholipase D1, prostaglandin E synthase, cyclooxygenase (Cox) 1, and Cox2 genes, but the expression of leukotriene synthesis enzyme 5-LOX was not affected. The superoxide dismutase (SOD)1, SOD2, nuclear factor erythroid-derived 2–like 2, and peroxiredoxin 1 genes, which are markers of reactive oxygen species (ROS), were also upregulated.

Schnabel et al. [63], after a 3-day culture of horse flexor digitorum superficialis tenocytes in five different culture media, observed that the presence of 100% PRP enhanced the expression of anabolic genes COL1A1, COL3A1 and COMP and elevated the COL1A1:COL3A1 ratio. However, no increase of catabolic MMP3 and MMP13 genes was detected and no correlation between PRP dosage and tenocyte metabolism was identified.

After 4 days of culture of the equine flexor digitorum superficialis tendon in 5 different matrices, McCarrel et al. [38] detected the higher expression of COL1A1, COL1A1:COL3A1 ratio, COMP and MMP3, and the decreased expression of COL3A1, decorin and MMP13 genes in PRP-containing media compared to controls.

Zhou et al. [66] cultured rabbit patellar tendon derived cells in matrices containing 10% LR-PRP or 10% PRP. At 14 days, LR-PRP was correlated with improved cell proliferation and MMP-1 MMP-13, IL-6, IL-1β, TNF-α, and PGE2 production, while PRP led to higher COL 1, COL 3 and α-smooth muscle actin gene expression. The authors also found no effect of PRP on non-tenocyte genes Sox-9, Runx-2 and peroxisome proliferator-activated receptor γ.

After culturing rat Achilles tendon cells on the PRP-collagen matrix, Xu et al. [64] found enhanced tenocyte viability, proliferation and migration potential compared to controls at 3 days. They also reported that PRP did not only simulate tendon-derived stem cells to form a tendinous fiber-like tissue but induced some tendon-derived stem cells to differentiate into vascular endothelial-like cells creating capillary tube-like structures. Additionally, and after 2 weeks, they noticed higher metabolic activity due to upregulation of the expression of collagen type I, collagen type III, SCX and TNC genes.

##### Human Studies

Anitua et al. [59], in 6-day cultures of human semitendinosus tenocytes, correlated the presence of PRP with accelerated cell proliferation and greater secretion of human procollagen type I C-peptide, compared to controls. Additionally, they found an increased synthesis of VEGF, TGF-β1 and HGF, which are molecules associated with neovascularization.

The same authors group also investigated the effect of PDGF and TGF-β1 growth factors, released from platelet α-granules, on PPP or PRP cultured tenocytes derived from human semitendinosus tendon. The addition of PDGF increased cell proliferation, while TGF-β1 was associated with higher collagen synthesis as well as stimulation of VEGF and HGF [58].

De Mos et al. [60] cultured tenocytes, derived from excised hamstring tendons of three children with a history of knee contracture, in media containing PPP or PRP at concentrations of 0%, 10% or 20%. They studied the effect of PRP on cell amount and morphology and the expression of genes associated with collagen metabolism and vascularization at 4, 7 and 14 days. On the 14th day, the tenocytes displayed a stretched, oblong shape compared to their spindle-shaped, fibroblast-like appearance at the beginning of the culture. Additionally, a dose-related correlation between their number and PRP was identified. Regarding growth factors, the expression of COL1 and COL3 genes was diminished by the 14th day, but the expression of MMP3, MMP13, VEGF-A and TGF-β1 was increased at the same time point. The MMP1 gene was upregulated until the 7th day, followed by a significant decrease until the 14th day. The authors concluded that these findings might be correlated with in vivo accelerated catabolic demarcation of injured tendons, angiogenesis, and the formation of fibrovascular callus.

Jo et al. [62], in 14-day cultures of tenocytes derived from degenerated rotator cuff, found that the addition of calcium-activated PRP stimulated cell proliferation and increased the synthesis of total collagen and glycosaminoglycans without provoking any morphological changes. Furthermore, compared to controls, PRP was associated with higher COL1 gene expression at day 7 but not at day 14, and increased COL3 gene expression at days 7 and 14 with a stable ratio COL3:COL1 at days 7 and 14. They also noticed higher decorin and SCX genes expression at day 14 and improved TNC gene expression at days 7 and 14.

Cross et al. [65] cultured supraspinatus tendon samples extracted from patients undergoing reverse shoulder arthroplasty for rotator cuff arthropathy. These chronic degenerative tendons were separated in two groups according to the histological severity of the degeneration. Tendon samples were cultured in matrices with or without LP-PRP or LR-PRP. In the moderate tendinopathy group, an increased COL1A1:COL3A1 ratio was found in PRP cultures and even moreso after LP-PRP injection. In the severe tendinopathy group, there was higher expression of MMP9 and IL-1β in media containing PRP. Finally, no impact of PRP on COMP and MMP13 expression was reported.

Zhang et al. [31] cultured human patellar tendon cells for 5 days and studied if proteinase-activated receptor (PAR)1-activated PRP and PAR4-activated PRP had different effects on tenocyte proliferation, cellular collagen production, morphology, gene expression, and differentiation compared to controls. Firstly, they found that the addition of PRP could increase the population doubling time and the collagen type I production. Secondly, the cells displayed a more elongated tenocyte-like appearance compared to controls. The PAR1-PRP and PAR4-PRP induced the differentiation of tendon stem/progenitor cells into tenocyte-like cells and promoted a further apparent “vessel-like” cellular pattern which was more evident and organized in the latter group. However, they showed different efficacy in the expression of COL1, MMP1 and MMP2 genes and did not significantly affect the expression of non-tenocyte-related genes (collagen type II, Runx2 and lipoprotein lipase).

#### 3.2.2. In Vivo Studies

The outcome of PRP on tendon metabolism and healing has been studied in sixteen studies. Either the direct injection of PRP at the site of the tendon tear or the application of PRP-collagen matrices on injured tendons were utilized [26,27,31,48,53,59,64,67,68,69,70,71,72,73,74,75]. The most examined tendons in animal models were the Achilles (*n* = 10) and patellar (*n* = 4) tendons [26,27,31,48,53,59,64,67,70,71,72,73,74,75]. The mean time of tendon histological examination after PRP application was 4 weeks (range: 5 days-23 weeks). Most studies were focused on histological findings that evaluated the collagen fiber density and orientation as well as the level of healing stage [31,48,53,59,64,67,68,70,71,72,73]. Treatment of tendon tear or inflammation with PRP was associated with the improvement of collagen fibers organization (*p* < 0.05) and significant reduction of healing time (*p* < 0.05). Furthermore, pro-inflammation macrophages were downregulated (*p* < 0.05) and COL 1 gene expression was increased (*p* < 0.05) in favor of PRP compared to controls (Table 4).

##### Animal Studies

Aspenberg and Virchenko [67] examined the effect of PRP injection on ruptured Achilles tendon in rats. On the 11th day of treatment, no obvious effect of PRP compared to placebo was detected. However, on the 21st day, a homogeneous mass of fibrous callus formation with a few inclusions of fat cells was visible in the PRP group, while a less mature healing structure was identified in controls.

Anitua et al. [59], a week after a course of four weekly saline, PPP or PRP injections in intact sheep Achilles tendons, observed that the tenocytes in both the PRP and PPP groups displayed an ovoid shape; they were aligned along the collagen fibers and they showed organization along the lines of tension. On the contrary, disordered and disorganized cells were accumulated in limited areas in the control group. Regarding the vasculature, greater vasodilatation, collapse and activation of endothelial cells were reported after PRP and PPP injections. In addition, higher cellular density within the fascicles was identified. The authors concluded that PRP injection within tendons was a safe and effective method that could enhance the tissue remodeling and healing processes.

In rabbit models, Lyras et al. [72] studied the effect of PRP on ruptured Achilles tendons. They observed a similar stage of healing after 1 and 2 weeks, more advanced tenocyte organization with less granular tissue at 3 weeks, and complete healing at 4 weeks in the PRP group compared to controls. Concerning the cluster of differentiation (CD) 31, which is a marker of vascular endothelial cells, significant higher values were identified at two weeks after PRP application. Afterwards, CD31 levels were sharply reduced and became completely absent at 4 weeks. The authors concluded that PRP could promote neovascularization, accelerate the healing process of injured tendons, and produce a better-quality scar tissue. In the course of the same experiment, they also studied the expression of IGF-I in the epitendon and endotendon. During the first 3 weeks, they found higher IGF-I expression in both epitendon and endotendon in the PRP group. On the contrary and by the 4th week, the control group showed a statistically significant overexpression of IGF-I in the endotendon but decreased expression in the epitendon [71].

Harris et al. [48] injected PRP on intact rabbit Achilles tendons and examined the histological findings after 2 and 6 weeks. To determine the boost effect of PRP, a group of rabbits was re-injected at week 6 and euthanized 12 weeks after the first injection. At 2 weeks, the authors noticed marked thickening of the peritenon with monocytic and lymphocytic inflammatory cells, areas of basophilic infiltration and vacuole formation, and the presence of multinucleated giant cells with new collagen bundles. At 6 weeks, chronic inflammation without any structural changes in the tendon tissue were noted. However, at 12 weeks, the inflammation was less predominant, and some calcified lesions were identified, probably due to reinjection.

After the transplantation of collagen-tendon-derived stem cells combined with PRP in ruptured rat Achilles tendons, Xu et al. [64] detected relatively normal tendon tissue after 3 weeks. Furthermore, the tenocytes showed a spindle-shaped morphology and relatively organized distribution along the longitudinal fibrous tissue of the tendon. Conversely, loose, thin, and poorly organized longitudinal fibrous tissue was identified when PRP was not utilized.

In a rabbit Achilles tendinopathy model created by intratendinous collagenase injection, Li et al. [70] studied the effect of leucocyte-rich-PRP on tissue healing. Concerning catabolic cytokines, early PRP injection (after 1 week) and late PRP injection (after 4 weeks) were associated with increased IL-10 and IL-6, respectively. No difference was found in IL-1β and TNF-α extraction between the two time points. Regarding gene expression, 1-week PRP was related with increased COL1, while 4-week PRP was related with higher COL3, MMP1 and MMP3 genes expression. Additionally, early PRP injection led to a more advanced healing process, as indicated by the parameters of fiber arrangement and structure, angiogenesis and nuclear and cell density. It was also associated with the increased number of CD163+ M2 macrophages and therefore decreased inflammation compared with the late PRP group. Based on all of these findings, the authors suggested the application of leucocyte-rich-PRP at the early stage of tendinopathy and not later.

Matsunaga et al. [53] used a PRP-fibrin matrix to bridge the gap of ruptured patellar tendons in a rabbit model. At 4 weeks, dense and longitudinally aligned collagen bundles were identified without any signs of immunological rejection.

Zhang et al. [73] specifically studied the effect of high-mobility group box (HMGB) 1 nuclear protein, which is released from platelets during injury, on tendon healing. In their experiment, they analyzed the healing process of ruptured patellar tendons in a two-mouse model, using either control or transgenic mice with a platelet-specific ablation of HMGB1. The authors applied three different solutions: saline, normal PRP, or PRP without the HMGB1 factor. At 7 days, normal PRP was correlated with faster tendon healing in both mice groups. In the control mice group, an almost typical tendon structure was obvious after normal PRP injection. In contrast to placebo and HMGB1-inhibited PRP treatment modalities, normal PRP was able to reduce the CD68+ M1 pro-inflammatory macrophages, regulate local inflammation, and increase CD146+ and CD73+ stem cells. This combination proved to be the most effective option regarding the recruitment of resident stem cells during tendon healing.

In another experiment in rats, Zhang et al. [31] studied the influence of PRP activated by thrombin or PAR1 or PAR4 on ruptured patellar tendons. A control group without PRP intervention was also examined. In 8 weeks, a completely healed tendon with the presence of overgrowth scar-like tissue was identified in the thrombin-PRP group. Furthermore, a tendon-like tissue with well-organized collagen fibers and very few blood vessels was apparent in the PAR4-PRP group, whereas unhealed tissue with some vessel-like structures were observed in both the PAR1-PRP and control groups.

Bosch et al. [68] examined the result of ultrasound-guided PRP or placebo injections on superficial digitorum tendon tears in horses. At 23 weeks, unidirectional arrangement of collagen fibrils with a fairly regular wave pattern and a more advanced structural arrangement were noticed after PRP injection compared to controls.

Yan et al. [75] studied the effect of LP-PRP or LR-PRP injections on rabbit Achilles tendon with chronic tendinopathy. At 4 weeks, the LR-PRP group demonstrated a better modified Movin score (grading system of tendinopathy severity) and more mature collagen fibers. Additionally, LR-PRP was related to decreased IL-6 production but showed no effect on IL-1β and TNF-α levels. The LP-PRP was also associated with increased COL1 and TIMP-1 and decreased MMP9 genes expression. Finally, both PRP groups were correlated with an increased COL1:COL3 ratio and decreased MMP1 and MMP3 genes expression, but did not influence the COL3 production.

In a similar rabbit Achilles tendon tendinopathy model, Jiang et al. [74] reported better modified Movin score and more mature collagen fibers at 6 weeks after PRP injection. Furthermore, at 3 weeks, LR-PRP was related to increased COL1, VEGF, VEGF receptor, TNF-a, arginase (ARG) 2, IL-10 and CD163 monoclonal antibody and decreased COL3 gene expression. At 6 weeks, LR-PRP was correlated with increased COL1 and CD163 and decreased COL3, VEGF, VEGF receptor, TNF-a, ARG2 and IL-10 gene expression, while LP-PRP showed higher ARG2 and IL-10 production.

Kobayashi et al. [26] investigated the influence of PRP hydrogel applied on rat patellar tendon full-thickness tear for a period of 10 weeks. Until the fourth week, PRP led to the earlier invasion of inflammatory cells, an increase in blood capillaries, a thickening of the tendon during the early phase of the tendon-repair process, and faster collagen rearrangement. Compared to controls, the PRP group showed a higher Bonar score (grading system of tendinopathy severity) at 2 and 4 weeks, but a lower one at 6 and 8 weeks, a lower ground substance score at 6 and 8 weeks, a higher vascularity score at 2 and 4 weeks, and lower collagen arrangement at 8 weeks.

Yu et al. [27] studied the effect of PRP on a rat Achilles tendon partial tear at 5 and 10 days after injection. After 5 days and compared to non-injected tendons, the PRP was associated with less random fibroblast orientation, more collagen matrix, further detected Ki-67–positive cells and less ED1+ macrophages. At 10 days, newly formed tendon fibers and abundant collagen fibers were found after PRP injection. However, the percentage of Ki-67–positive cells and ED1+ macrophages was similar in both groups. The application of PRP was also linked to decreased cell apoptosis at both timepoints.

Han et al. [69] examined, the result of injected PRP or mesenchymal stem cells or both on growth factors secretion, osteogenic differentiation ability and cell death resistance in a rat rotator cuff repair model. Platelet-ich plasma was found to enhance the secretion of VEGF, PDGF, EGF, TGF-β, BMP2 and BMP7 growth factors that could promote recovery through angiogenesis, bone formation and tissue regeneration. Moreover, the increased expression of COL1, TNMD, SCX and P-ERK1/2 was noticed after PRP therapy. Regarding osteogenic differentiation, PRP enhanced the expression of Osterix, Runx2 and osteocalcin compared to controls. The authors concluded that the expression of all these genes was even higher when a combination of PRP and mesenchymal stem cells was utilized.

## 4. Limitations

This systematic review has inherent limitations that are mostly related to the type of published studies. Specifically, there was inconsistency among the studies with respect to the methods of preparation and the concentration of platelets and leukocytes in PRP. Secondly, in vivo studies showed a significant variation regarding the time of culture and duration of study period. Finally, the laboratory-induced pathologies of ligaments and tendons after the application of PRP may differ from the traumatic or chronic conditions existing in clinical practice.

## 5. Conclusions

This review confirmed the positive molecular effect of PRP on ligament and tendon injury and pathology. In vitro and in vivo animal and human studies focused on tissue healing and remodeling and examined the histological changes and the upregulation of the release of anabolic growth factors and gene expression. The application of PRP has proved to stimulate fibroblast and tenocyte proliferation and migration to the affected site and to increase the production of collagen with proper organization. This process is amplified by the upregulation of collagen type I gene expression, which leads to healing tissue with better biomechanical properties. Notably, the biggest impact of PRP was observed on cells from immature donors compared to adults’ cells. On the contrary, other studies demonstrated that PRP had no effect or even a negative effect on tendon and ligament regeneration. These controversies are mainly related to the variable processes and methodologies of the preparation of PRP, necessitating standardized protocols for both investigation and application.

## Figures and Tables

**Figure 1 ijms-24-02744-f001:**
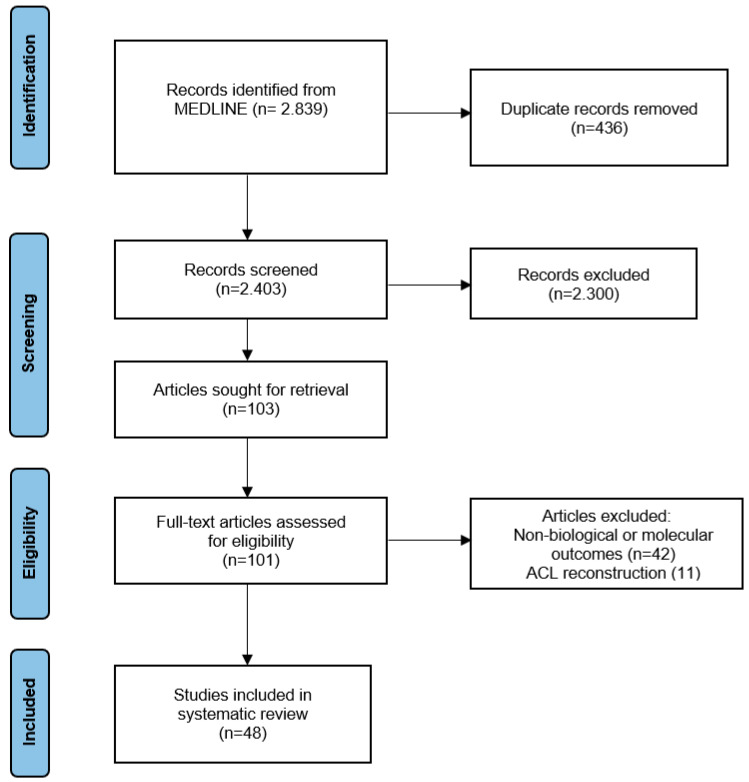
Flow chart diagram for inclusion and exclusion article process.

**Table 1 ijms-24-02744-t001:** Histological and molecular effect of PRP on ligaments in in vitro studies.

Article	Year	Animal (Species)/Human	Number of Animals or Participants	Ligaments	Solutions Compared	Duration of Culture	Histology and Cellular Activity	Molecular Effect
Smith et al. [41]	2006	Animal (horse)	5	Suspensory ligament	control, equine serum, foetal bovine serum, ABM (5%, 10%), PRP (5%, 10%)	48 h		↑ COMP, 3H-Leucine incorporation
Schnabel et al. [40]	2008	Animal (horse)	6	Suspensory ligament	control, whole blood, BMA, PPP, PRP (10%, 50% or 100%)	3 days		↑ COMP, MMP3 ↓ COL3A1, MMP13 ≈ COL1A1, COL1A1:COL3A1, Decorin
McCarrel et al. [38]	2009	Animal (horse)	5	Suspensory ligament	control, PPP, PRP, BMA	96 h		↑ COL1A1, COMP, Decorin, MMP3, COL1A1:COL3A1 ↓ COL3A1, MMP13
Cheng et al. [34]	2010	Animal (pig)	Nm	ACL	control, platelet, PPP, PRP	14 days	↑ metabolic activity, collagen organization ↓ apoptotic rate Centrally positioned elongated nuclei, oriented along the longitudinal axis; highest average nuclear aspect ratio; wave fiber-like structure of collagenous ECM aligned with the longitudinal axis of the constructs; prevalence of collagen fibrils, mostly aligned with the cells and packed into collagen fibers	↑ COL1A1, COL3A1
Cheng et al. [33]	2012	Animal (pig)	15	ACL	control, PRP	14 days	↑ metabolic activity, collagen organization ↓apoptotic rate (more in immature and adolescent animals)	↑ COL1A1, COL3A1 (more in immature and adolescent animals)
Yoshida et al. [44]	2013	Animal (pig)	Nm	ACL	control, PPP, PRP, PBMC, PPP + PBMC, PRP + PBMC	14 days	PRP + PBMC ↑ metabolic activity, collagen	PRP + PBMC ↑ procollagen type I and type III genes, IL-6
Yoshida et al. [42]	2014	Animal (pig)	5	ACL	control, PPP, PRP, PRPx3, PRPx5	14 days	↑ metabolic activity, viability	↑ COL1A1, COL3A1
Zheng et al. [43]	2020	Animal (rabbit)	4	ACL	control, Sanguisorba officinalis L. polysaccharide + PRP, PRP	72 h	↑ viability, migration ↓ apoptotic rate	↑ Runx2, ALP, BMP2, COL1, OPG ↓ TLR-4, p65 phosphorylation
Fallouh et al. [36]	2010	Human	4	ACL	control, PRP, PPP 5%, PRP 10%	7 days	↑ viability	↑ total collagen, COL3 ≈ COL1, collagen/μg DNA
Anitua et al. [32]	2013	Human	3	Periodontal ligament	control, PRP	24, 48, 72 h	↑ proliferation, migration, attachment to COL1	↑ VEGF, TSP-1, HGF, Procollagen I, CTGF ↓ α2-Integrin
Dhillon et al. [35]	2015	Human	11	ACL	control, 5% PPP, 5% PRP, 10% PRP	2 days	↑ viability, proliferation ≈ apoptosis	
Krismer et al. [37]	2017	Human	5	ACL	controls, 2.5% LR-PRP, 2.5% PRP, 20% LR-PRP	7, 14, 21 days	↑ proliferation ≈ extracellular matrix production	↑ COL1A2, COL2A1, TNC, SCXA, TNMD, ACAN, MKX, MMP13 ↓ COL3A1, MMP3
Rattanasuwan et al. [39]	2018	Human	3	Periodontal ligament	control, 5% PRP, 10% PRP	6 h	↑ proliferation, migration, attachment to COL1 PRP (5% and 10%) Increased periodontal ligament fibroblasts cells with spindle shape and positive stain for toluidine blue O	

Abbreviations: ↑: Increase, ↓: Decrease, ≈: Equal, ACAN: Aggrecan, ACL: Anterior cruciate ligament, ALP: Alkaline phosphatase, BMA: Bone marrow aspirate, BMP: Bone morphogenetic protein, COL: Collagen, COMP: Cartilage oligomeric matrix protein, CTGF: Connective tissue growth factor, ECM: Extra-cellular matrix, HGF: Hepatocyte growth factor, IL: Interleukin, LR-PRP: Leukocyte rich platelet rich plasma, MKX: Mohawk homeobox, MMP: Matrix metalloproteinase, OPG: Osteoprotegerin, PBMC: Peripheral blood mononuclear cells, PPP: Platelet poor plasma, PRP: Platelet rich plasma, Runx2: Runt-related transcription factor 2, SCXA: Scleraxis bHLH transcription factor, TLR-4: Toll-like Receptor 4, TNC: Tenascin C, TNMD: Tenomodulin, TSP-1: Thrombospondin-1, VEGF: Vascular endothelial growth factor.

**Table 2 ijms-24-02744-t002:** Histological and molecular effect of PRP on ligaments in in vivo animal studies.

Article	Year	Animal Species	Number of Animals	Ligaments (Pathology)	Comparison Modalities	Time from PRP Application	Histology and Cellular Activity
Murray et al. [56]	2006	Dog	12	ACL (tear)	control, collagen-PRP composite	3,6 weeks	↑ defect filling
Murray et al. [54]	2007	Pig	5	ACL (tear)	control, collagen-PRP composite	4 weeks	↑ cellularity, vascularity ↓ inflammation
Murray et al. [55]	2007	Dog	17	ACL, MCL, Patellar tendon (full thickness tear)	extraarticular ligament, collagen-PRP hydrogel at ACL	3, 6 weeks	≈ modified LTMI to extraarticular ligaments
Joshi et al. [50]	2009	Pig	27	ACL (full thickness tear)	control, collagen-PRP matrix	4,6 weeks, 3 months	↑ cellularity ≈ vascularity Increased number of fusiform-shaped cells
Mastrangelo et al. [52]	2011	Pig	8	ACL (full thickness tear)	collagen-3xPRP matrix, collagen-5xPRP matrix	13 weeks	5xPRP ↑ modified LTMI Similar cell orientation and shape in both groups
Harris et al. [48]	2012	Rabbit	18	MCL (full thickness tear)	control, PRP	2,6 weeks, 12 weeks (reinjection at 6 weeks)	↓ prominent inflammation at 6 and 12 weeks
Haus et al. [49]	2012	Pig	24	ACL (full thickness tear)	control, collagen-PRP composite	15 weeks	↑ remodeling at the insertion site, cellularity, collagen organization (less in older age animals) Increased collagen organization at fibrous zone with relatively large proportion of collagen oriented perpendicular to the insertion site in a densely packed arrangement; relatively good organization of collagen perpendicular to the subchondral plate at fibrocartilage zone; distinct fibrocartilage layer
Matsunaga et al. [53]	2013	Rabbit	20	MCL (full thickness tear)	control, PRP-fibrin scaffold	4, 8, 12 weeks	↑ mature neo-ligament at 12 weeks Neo-ligament of dense and longitudinally aligned collagen bundles and direct fibrocartilage insertions
Yoshika et al. [57]	2013	Rabbit	31	MCL (full thickness tear)	control, clot of PRP	3, 6 weeks	↑ collagen organization ≈ cellular size, cellularity
Amar et al. [45]	2015	Rat	32	MCL (full thickness tear)	control, PRP	3 weeks	≈ cellularity, collagen organization, vascularityFat cells; inflammatory foci; loose and disorganized collagen
Bozynski et al. [46]	2016	Dog	12	ACL (partial tear)	standard care, washout, LP-PRP		≈ myxoid/mucinous degeneration, collagen fiber orientation
Cook et al. [47]	2016	Dog	12	ACL (partial tear)	control, LP-PRP (inj. at 1, 2, 3, 6 and 8 weeks)	24 weeks	↓ severe pathology
LaPrade et al. [51]	2018	Rabbit	80	MCL (full thickness tear)	control, PPP, 2xPRP, 4xPRP, intact	6 weeks	≈ LTMI to controls ↓ LTMI than intact ligament Negative effect of 4xPRP

Abbreviations: ↑: Increase, ↓: Decrease, ≈: Equal, ACL: Anterior cruciate ligament, FGF: Fibroblast growth factor, LP-PRP: Leukocyte poor platelet rich plasma, LTMI: Ligament tissue maturity index, MCL: Medial collateral ligament, PDGF: Platelet-derived growth factor, PRP: Platelet rich plasma, TGF: Transforming growth factor, vWF: von Willebrand factor.

**Table 3 ijms-24-02744-t003:** Histological and molecular effect of PRP on tendons in in vitro studies.

Article	Year	Animal (Species)/Human	Number of Animals or Participants	Tendon	Solutions Compared	Duration of Culture	Histology and Cellular Activity	Molecular Effect
Schnabel et al. [63]	2007	Animal (horse)	6	Flexor digitorum superficialis	control, whole blood, BMA, PPP, PRP (10%, 50% or 100%)	3 days		↑ COL1A1, COL3A1, COL1A1:COL3A1, COMP ↓ MMP13 ≈ MMP3, Decorin
McCarrel et al. [38]	2009	Animal (horse)	5	Flexor digitorum superficialis	control, 100% BMA, 100% PRP, lyophilized platelet product	96 h		↑ COL1A1, COL1A1:COL3A1, COMP, MMP3 ↓ COL3A1, Decorin, MMP13
Zhou et al. [66]	2015	Animal (rabbit)	2	Patellar	control, 10% LR-PRP, 10% PRP	14 days	LR-PRP ↑ cell proliferation	PRP ↑ COL1, COL3, α-SMA LR-PRP ↑ MMP1, MMP13, IL-6, IL-1β, TNF-α, PGE_2_
Hudgens et al. [61]	2016	Animal (rat)	Nm	Tail tendon	control, PPP, PRP	5 days		↑ BMP7, CCL2, CCL7, IL-1a, IL-6, IL-10, TNFa, LOX, COL8, Lubricin, MMP3, MMP9, MMP10, MMP13, Ki67, Fosb, Fosl1, c-Jun, PLD1, PTGES, Cox1, Cox2, SOD1, SOD2, NFE2L2, Prdx1 ↓ CTGF, IGF1, IL-15, Elastin, COL1, COL3, CILP, Fibromodulin, COL12, COL14, EGR1, EGR2, SCX, TNMD ≈ TGFβ, IL-1b, VEGF, HAS1, HAS2, MMP8, MMP2, TIMP1, TIMP2, Atg10, Bnip1, GABARAPL2, beclin 1, Trim13, SIRT1, iNOS, 5-LOX
Xu et al. [64]	2017	Animal (rat)	Nm	Achilles	control, 10% PRP	3 weeks	↑ viability, migration, proliferation at 3 days	↑ COL1, COL3, SCX, Tenascin-C
Anitua et al. [59]	2006	Human	6	Semitendinosus	control, PPP, PRP	6 days	↑ proliferation	↑ VEGF, TGF-β1, HGF, human Procollagen I C-peptide
Anitua et al. [58]	2007	Human	4	Semitendinosus	control, PPP or PRP with or without PDGF or TGF-β1	4 days	PDGF ↑ proliferation TGF-β1 ↑ collagen synthesis	TGF-β1 group ↑ VEGF, HGF
de Mos et al. [60]	2008	Human	3	Hamstring	control, PRP or PPP (10% or 20%)	4, 7, 14 days	↑ proliferation, collagen production Changes in cells appearance: from spindle-shaped, fibroblast-like cells to stretched, oblong shaped cells	↑ MMP3, MMP13, VEGF-A, TGF-β1 ↓ COL1, COL3, MMP1. ≈ COL3:COL1
Jo et al. [62]	2012	Human	9	Rotator cuff	control, PPP, PRP, PRP + thrombin	14 days	↑ proliferation, total collagen, glycosaminoglycans	Day 7 ↑ COL1, COL3, Tenascin-C ≈ COL3:COL1 Day 14 ↑ COL3, Decorin, SCX, Tenascin-C ≈ COL3:COL1
Cross et al. [65]	2015	Human	20	Supraspinatous	control, LP-PRP, LR-PRP			↑ COL1:COL3, MMP9, IL-1b ≈ COMP, MMP13
Zhang et al. [31]	2019	Human	7	Patellar	control, PRP + thrombin, PRP + PAR1, PRP + PAR4	5 days	↑ proliferation PRP + thrombin Elongated tenocyte-like cells PRP + PAR1 Tenocyte-like cells, “vessel-like” cellular pattern PRP + PAR4 More organized cells than PRP + PAR1 group	↑ COL1, MMP1, MMP2 ≈ COL2, Runx-2, LPL

Abbreviations: ↑: Increase, ↓: Decrease, ≈: Equal, α-SMA: α-Smooth muscle actin, Atg: Autophagy-related protein, BMA: Bone marrow aspirate, BMP: Bone morphogenetic protein, Bnip: B-cell lymphoma/adenovirus E1B interacting protein, c-JUN: Transcription factor Jun, CCL: Chemokine (C-C motif) ligand, CILP: Cartilage intermediate layer protein, COL: Collagen, COL1A1: Collagen type I alpha 1 chain, COL3A1: Collagen type III alpha 1 chain, COMP: Cartilage oligomeric matrix protein, Cox: Cyclooxygenase, CTGF: Connective tissue growth factor, Fosl1: FOS-related antigen 1, GABARAPL2: Gamma-aminobutyric acid receptor-associated protein–like 2, HAS: Hyaluronan synthase, HGF: Hepatocyte growth factor, IGF: Insulin-like growth factor, IL: Interleukin, iNOS: Inducible nitric oxide synthase, LOX: Lipoxygenase, LPL: Lipoprotein lipase, LP-PRP: Leukocyte poor platelet rich plasma, LR-PRP: Leukocyte rich platelet rich plasma, MMP: Matrix metalloproteinase, NFE2L: Nuclear factor (erythroid-derived 2)–like, PAR: Proteinase-activated receptor, PLD1: Phospholipase D1, PPP: Platelet poor plasma, Prdx: Peroxiredoxin, PRP: Platelet rich plasma, PTGES: Prostaglandin E synthase, Runx2: Runt-related transcription factor 2, SCX: Scleraxis, SIRT: Sirtuin, SOD: Superoxide dismutase, TGF: Transforming growth factor, TNFa: Tumor necrosis factor alpha, TNMD: Tenomodulin, Trim: Tripartite motif containing, VEGF: Vascular endothelial growth factor.

**Table 4 ijms-24-02744-t004:** Histological and molecular effect of PRP on tendons in in vivo animal studies.

Article	Year	Animal Species	Number of Animals	Tendons	Comparison Modalities	Time from PRP Application	Histology and Cellular Activity	Molecular Effect
Aspenberg and Virchenko [67]	2004	Rat	80	Achilles (full thickness tear)	control, PPP, PRP	11, 21 days	21 days ↓ time to healing	
Anitua et al. [59]	2006	Sheep	6	Achilles (intact)	control, PPP or PRP (4 inj/1 per week)	7 days after last injection	↑ cell density, vascularity	
Lyras et al. [72]	2009	Rabbit	48	Achilles (full thickness tear)	control, PRP	1, 2, 3, 4 weeks	↓ time to healing	Week 1,2 ↑ CD31 Week 3,4 ↓ CD31
Lyras et al. [71]	2011	Rabbit	48	Achilles (full thickness tear)	control, PRP	1, 2, 3, 4 weeks	↓ time to healing	Until 3 weeks ↑ IGF-I in epitendon and endotendon Week 4 ↑ IGF-I in epitendon ↓ IGF-I in endotendon
Bosch et al. [68]	2011	Horse	6	Superficial digitorum flexor (full thickness tear)	control, PRP	23 weeks	↑ collagen organization	
Harris et al. [48]	2012	Rabbit	18	Achilles (intact)	control, PRP	2, 6 weeks, 12 weeks (reinjection at 6 weeks)	↓ inflammatory response	
Matsunaga et al. [53]	2013	Rabbit	18	Patellar (full thickness tear)	control, PRP-fibrin scaffold	4 weeks	↑ collagen organization	
Xu et al. [64]	2017	Rat	45	Achilles (full thickness tear)	control, collagen-matrix, PRP-collagen matrix	1, 2, 3 weeks	↑ cell maturation, collagen organization	
Yan et al. [75]	2017	Rabbit	28	Achilles (chronic tendinopathy)	control, LP-PRP, LR-PRP	4 weeks	LR-PRP ↑ modified Movin score, mature collagen fibers	LR-PRP ↓ IL-6 ≈ IL-1β, TNF-α. LP-PRP ↑ COL1, TIMP-1, ↓ MMP-9. LR-PRP and LP-PRP ↑ COL1:COL3 ↓ MMP1, MMP3. ≈ COL 3
Han et al. [69]	2019	Rat	Nm	Rotator cuff (full thickness tear)	control, MSC, PRP, PRP + MSC	4 weeks		↑ VEGF, PDGF, EGF, TGF-β, BMP2, BMP7, COL1, TNMD, SCX, *p*-ERK1/2, Osterix, Runx2, OCN
Zhang et al. [31]	2019	Rat	8	Patellar (partial tear)	control, PRP + thrombin, PRP + PAR1, PRP + PAR4	8 weeks	↓ time to healing (faster healing when thrombin-activated)	
Jiang et al. [74]	2020	Rabbit	28	Achilles (tendinopathy)	control, LP-PRP, LR-PRP	3, 6 week	LR-PRP ↑ modified Movin score, mature collagen fibers	Week 3—LR-PRP ↑ COL1, VEGF, VEGF receptor, TNF-a, ARG2, IL-10, CD163 ↓ COL3 Week 6—LR-PRP ↑ COL1, CD163 ↓ COL3, VEGF, VEGF receptor, TNF-a, ARG2, IL-10 Week 6—LP-PRP ↑ ARG2, IL-10
Kobayashi et al. [26]	2020	Rat	40	Patellar (full thickness tear)—	control, PRP gel	2, 4, 6, 8, 10 weeks	↓ time to healing ↑ collagen rearrangement, vascularity, tendon thickening Week 2,4 ↑ Bonar score, vascularity score Week 6,8 ↓ Bonar score, ground substance score, collagen arrangement PRP Earlier invasion of inflammatory cells; increase of blood capillaries; thickening of tendon during the early phase; collagen rearrangement	
Li et al. [70]	2020	Rabbit	32	Achilles (tendinopathy)	control, LR-PRP (at 1 or 4 week)	6 weeks	↑ collagen organization, vascularity, cell density	Week 1 ↑ IL-10, COL1, CD163+ M2 macrophages Week 4 ↑ IL-6, COL3, MMP1, MMP3 ≈ IL-1β, TNF-α
Yu et al. [27]	2021	Rat	48	Achilles (partial tear)	control, PRP	5, 10 days	Days 5 ↑ fibroblast orientation, collagen matrix ↓ cell apoptosis Day 10 ↑ newly formed tendon fibers, collagen fibers	Days 5 ↑ Ki-67–positive cells ↓ ED1+ macrophages Day 10 ≈ Ki-67–positive cells, ED1+ macrophages
Zhang et al. [73]	2021	Mouse	18	Patellar (partial tear)	control, PRP with or without HMGB1	7 days	↓ time to healing (faster healing when HMGB1 present)	↑ CD146+ and CD73+ stem cells ↓ CD68+ M1 macrophages

Abbreviations: ↑: Increase, ↓: Decrease, ≈: Equal, ARG: Arginase, BMP: Bone morphogenetic protein, CD: Cluster of differentiation, COL: Collagen, EGF: Epidermal growth factor, HMGB: High-mobility group box, IGF: Insulin-like growth factor, IL: Interleukin, LP-PRP: Leukocyte poor platelet rich plasma, LR-PRP: Leukocyte rich platelet rich plasma, MMP: Matrix metalloproteinase, MSC: Mesenchymal stem cells, OCN: Osteocalcin, PAR: Proteinase-activated receptor, PDGF: Platelet-derived growth factor, PPP: Platelet poor plasma, PRP: Platelet rich plasma, Runx2: Runt-related transcription factor 2, SCX: Scleraxis, TIMP: Tissue inhibitor of metalloproteinases, TGF: Transforming growth factor, TNF-a: Tumor necrosis factor alpha, TNMD: Tenomodulin VEGF: Vascular endothelial growth factor.

## Data Availability

Not applicable.

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
