# Peer review of "Molecular and Biologic Effects of Platelet-Rich Plasma (PRP) in Ligament and Tendon Healing and Regeneration: A Systematic Review"

_ijms, 2023, doi:10.3390/ijms24032744_

Round 1

Reviewer 1 Report

Treatment of diseases of the tendon-ligamentous apparatus is a serious problem not only for medicine, but is also relevant for veterinary medicine, especially equestrian sports. Therefore, a thorough review of the biological effect of platelet-rich plasma (PRP) on the regeneration of ligaments and tendons, as well as the study of the molecular mechanisms of such regulation, is of theoretical and practical interest. The authors systematized the information presented in 46 basic publications on this topic, but the analysis of the molecular mechanisms of regeneration of damaged ligaments is not sufficiently covered. It is also necessary to decipher many abbreviations that are understandable only to a narrow circle of specialists. The use of Plasmolifting technology is widely practiced in veterinary clinics in Russia. At the same time, experts say that the method of PRP administration and subsequent therapy is very important for obtaining a positive result, which is desirable to reflect in the discussion of the results.

Reviewer 2 Report

Review of IJMS 2092571

 Molecular and Biologic Effects of platelet-rich plasma (PRP) in ligament and tendon 2 healing and regeneration. A systematic review

This is an interesting review however its English needs improvement, Table contents need to be improved and the general organization of the manuscript needs improvement.  I have made a few suggestions on how I think your review could be significantly improved and expanded in relevant areas.

What are the bracketed numbers after author names in the title about ?

The “Affiliation 1 and 2” need to be removed, the superscript number specifies this information.  Do the authors have ORCID Numbers?  Does the corresponding author have an institutional e-mail?

The bullet points on lines 128, 170, 207, 267, 306, 359 are not required and do not comply to journal format.

A discussion sub-heading in a review is inappropriate since the entire manuscript is a discussion of the literature.

Line 736 spelling should be phosphatase

 Line 36 The authors state “Once activated, platelets release many growth factors that enhance healing of the injured tissues [3]”.  Could the authors list the growth factors involved in these healing responses and their particular attributes.

Line 65 “variables using z-test and Bonferroni correction was used to adjust for p values. All tests were two-sided and statistical significance was assumed at a p value of less than 0.05.”  P statistical values should be P CAP italic

The authors need to provide some information on the composition of tendons/ligaments and their functional components and how these are altered with degeneration and in repair processes.  What about the specific roles of particular collagens and tendon proteoglycans and elastin in tendon and ligament repair.

Brief comments on the role of biomechanical stresses on tendon/ligament repair processes also need to be covered.  This is a major omission given the functional roles of tendons/ligaments and the responsiveness of tendon/ligament cells to these stresses.

Line 118 The authors make the comment “Thirteen studies examined the correlation of PRP with the upregulation of ligament anabolic activity using incubating cultures of ligament fibroblasts in PRP-containing matrices [18-30]. “ Can the authors comment on the ECM components that were stimulated by PRP treatment in tendon repair and the growth factors involved.

Table 1  Cheng entry  2010 – can the authors correct  the spelling for platelet

In Table 1 the information provided in the Histology column needs to be more informative for all studies cited. What ECM components actually change? Is their organization/assembly altered?  How does this effect their functional properties ie ability to resist tension and maintainenance of the strength of the temndon.  Only information relevant to histology should be covered in this column.  The authors need to distinguish between the properties of tendons and  ligaments.  Tendons are essentially inextensible and are designed for force transmission to muscle.  Ligaments link bone to bone and have an appreciably higher elastin content than tendon.

A recent study in an ACL rupture model in rats showed that if knee kinematics were controlled appropriately to minimise abnormal tibial translation then spontaneous repair of the transected ACL occurred and recovery of 50% of the ACL biomechanical properties was achieved in an 8 week recovery period (Kokubun, 2016).  Since platelets are primary responders in spontaneous repair processes I wonder if the authors might want to comment on the role of mechanical stabilization in tendon repair processes to provide insights into how tendon repair might be improved.

I can understand why the authors did not cover this area due to the exclusion criteria they set in the introduction however this is nevertheless an important central issue in the providision of a better understanding of tendon repair responses.

Kokubun T, Kanemura, N, Murata, K, Moriyama, H, Morita, S, Jinno, T, Ihara, H, Takayanagi, K. (2016) Effect of Changing the Joint Kinematics of Knees With a Ruptured Anterior Cruciate Ligament on the Molecular Biological Responses and Spontaneous Healing in a Rat Model. Am J Sports Med, 44:2900-2910.

Table 2  The molecular effect column is mainly incomplete, all details in each category need to be provided or the column deleted-as it stands it provides very little information which could be easily covered in a sentence in the text.

Table 3 information provided in the Histology column is incomplete.

In the Tables a listing of the abbreviation information would be useful in a footnote to each Table.

The conclusions section needs to be re-worded and written in an incisive informative manner.  The reader needs to obtain an important take-home message from this section and be interested enough to delve into the text to get further information.

English use

Lines 128-253 The English construction needs improvement throughout.

Tables

It is unclear what numbers of participants means-is this the number of replicates?

Why were no histology plates illustrating normal tendons and those repaired by PRP made available in this review? these would have been highly informative to the reader.  The “histology” information provided in the Tables in your review is not particularly informative.

See Zhang J, Nie D, Williamson K, Rocha JL, Hogan MV, Wang JH. Selectively activated PRP exerts differential effects on tendon stem/progenitor cells and tendon healing. J Tissue Eng. 2019 Jan 16;10:2041731418820034.

You will of course need to obtain permission to re-use this material if you decide to use it in your review, however this is a straightforward process conducted by the copyright transfer office. Details of this are on the entry page of this publication in the journal under “request permissions”.  This paper also covers an area that you have not covered in your review namely tenocyte progenitor stem cells and tendon repair.   This will also improve the coverage of your review and an additional paragraph in the text should be included in your revision.

Line 208 The authors make the comment “Murray et al [41] found that extraarticular ligaments healed faster than intraarticular ones after studying MCL, patellar tendon and ACL tears in dog models. After applying a collagen-PRP hydrogel at the ligament tear, they compared the healing process between the intraarticular ACL and the extraarticular MCL ligaments after 6 weeks. No difference in LTMI score and similar levels of fibronectin, fibrinogen, PDGF-A, TGF-β1, FGF-2, procollagen type I and vWF growth factors were reported, indicating decrease healing time of the intraarticular ACL over the extraarticular ligament section”.  Biomechanical stresses are important determinants that effect tendon healing-yet there is no apparent appreciation of this here, this needs to be rectified. See also my earlier comment on the Kokobun ACL study in rats, tendon stabilization and spontaneous repair.

The authors need to comment on the variability of PRP preparations and methodology for preparation of personalized PRP fractions in clinical practice and practical limitations.  As to whether PRP is efficacious or not has been a controversial area and is probably related to the highly variable results reported for this methodology due to the highly variable composition of the PRP preparations that have been used.  As far as I am aware standardized methodologies have not been developed so far for PRP therapy and this needs to be done to move this therapeutic area forward.  This is also an area the authors could discuss further and is clearly relevant to the content of this review.

Reviewer 3 Report

More figures are needed. It is very difficult to follow the text with no figures.

Reviewer 4 Report

It is an interesting systematic review on platelet-rich plasma used for treatment of ligament and tendon pathologic conditions. Paper is well-organized and can be easily followed. I have only some minor questions before it can be accepted. 1) affiliation should be corrected

2) inclusive and exclusive criterion - can author provide more detailed reason why so many papers have been excluded? i.e. 2300 paper excluded but without giving clear reason.

Once author clearly state why so many papers have been excluded within the present study then work would be worth being published.

Round 2

Reviewer 2 Report

Review of IJMS 2092571: Molecular and Biologic Effects of platelet-rich plasma (PRP) in ligament and tendon healing and regeneration. A systematic review

The revised version of the manuscript is considerably improved.  I only noticed a few very minor items that need correction.

The orcid numbers should not be in the title these need to be identified as such with the ORCID logo as per journal format and listed after the acknowledgements – the copy editor can advise further on this.

Some entries in Table I need correction by correctly aligning contents of the columns. See Smith, Cheng, Krismer entries

Accepted standardised abbreviations of journal names should be used in the reference list.

The same text type should be used throughout the manuscript.

Reviewer 3 Report

Accept as it is.
